# Impact of a Brief Educational Intervention on Knowledge, Perceived Knowledge, Perceived Safety, and Resilience of the Public During COVID-19 Crisis

**DOI:** 10.3390/ijerph17165971

**Published:** 2020-08-17

**Authors:** Arielle Kaim, Eli Jaffe, Maya Siman-Tov, Ella Khairish, Bruria Adini

**Affiliations:** 1Department of Emergency Management and Disaster Medicine, School of Public Health, Sackler Faculty of Medicine, Tel Aviv University, P.O. Box 39040, Tel Aviv 6139001, Israel; ariellekaim@mail.tau.ac.il; 2PR, Training and Volunteers division, Magen David Adom, Igal Alon 70 6706215 Tel Aviv, Israel; Eliy@mda.org.il (E.J.); maylev90@gmail.com (M.S.-T.); EllaK@mda.org.il (E.K.); 3Department of Emergency Medicine, Ben Gurion University of the Negev, P.O. Box 653, Beer Sheva 8410501, Israel

**Keywords:** COVID-19, knowledge, trust, practices, educational intervention, resilience, pandemic

## Abstract

Extraordinary and unprecedented public health measures have been implemented to contain the ongoing spread of the coronavirus disease 2019 (COVID-19) pandemic. There is paramount importance of cooperation and population engagement in reducing disease infection rates and relieving an outbreak’s burden on society. The civil society’s engagement may be achieved through disaster education interventions. In this cross-sectional study, a pre-post questionnaire was used to investigate the impact of a brief educational intervention on knowledge, perceived knowledge, perceived safety, and the individual resilience of the population relating to the COVID-19 outbreak. The results of the study display the benefits of the educational intervention to include a significant overall increase in all examined variables. The study also reviewed the overall trust of the public concerning the main responding authorities, as well as practices concerning protective measures for COVID-19. This study demonstrates that educational interventions, such as the brief video, provide an easily implementable design and effective means for educating and empowering the public and should, thus, be considered as a component of future outbreak responses.

## 1. Introduction

Coronavirus disease 2019 (COVID-19) is an emerging viral respiratory disease that originated, and was initially detected, in Wuhan, China, in December 2019. This novel illness is characterized by its high transmissibility and clinical symptoms of varying severity, most commonly including fever, cough, fatigue, and myalgia [1]. Furthermore, the disease has a distinct epidemiological signature, whereby it causes disproportionate frequency of critical illness in older patients and those with underlying comorbidities [2,3,4]. From the onset of its outbreak, COVID-19 has spread to 188 countries and regions, with over 5.2 million (5,260,624) laboratory-confirmed cases and over 339 thousand (339,627) deaths as of May 23, 2020 [5]. The World Health Organization (WHO) declared the outbreak a pandemic on March 11, 2020 [6].

In response to the global increase of reported cases and deaths, the outbreak has resulted in the implementation of many extraordinary and unprecedented public health measures to reduce further spread of the virus. To inform these efforts, the world has looked to past successes and failures of managing infectious disease outbreaks. In particular, these included SARS-CoV in 2002 and Middle East respiratory syndrome (MERS)-CoV in 2012, both of which were caused by coronaviruses [7,8,9]. One such lesson indicates the paramount importance of cooperation and population engagement in reducing disease infection rates and relieving an outbreak’s burden on society [10,11,12]. The civil society’s engagement may be achieved through disaster education interventions [13]. Previous reviews of educational interventions’ efficacy have revealed the successful use of telephone health education on improving knowledge and practices during SARS-CoV [14]. An additional study that assessed the efficacy of a video intervention concluded that the intervention positively impacted health beliefs but did not improve health behavior [15]. As seen in Saudi Arabia during the MERS-CoV, social networking site interventions were shown to improve health behavior-related outcomes [16].

In the context of disaster preparedness, knowledge may be defined as “the practice of intervening purposefully” to elucidate appropriate function, while perceived knowledge is the belief of oneself concerning the ability to implement such a function [17]. Findings indicate that the public’s knowledge is thought to have significant influence on its uptake of and adherence to control measures; conversely, knowledge gaps among the population can contribute to the reduced efficacy of control measures implemented [18,19]. Paton, Mclure and Burgelt (2006) describe that perceived knowledge is often an overestimation and discrepant with actual levels. Overestimations of knowledge are linked to increased perception of safety (believing oneself is protected, regardless of the actual situation) and a reduced perception of risk (judgement of the severity of a hazard) [20,21], resulting in poor mitigation. In contrast, high-risk perception prompts greater community preparedness and resilience [22]. Opposing findings were established in the context of climate change, where actual knowledge corroborates people’s perception of knowledge [23]. These conclusions suggest that higher individual perceptions of knowledge can enhance community resilience.

An additional element that impacts the decision-making process for uptake of control measures is perceived individual resilience, which has been associated with increased education and awareness [24]. Personal resilience is defined as “the ability to encounter and move through significant hardship” [25]. High-perceived resilience has been correlated with catalyzing positive behavioral engagement in disaster preparedness [26]. Thus, to facilitate and ensure proper management of the current crisis and future outbreaks, there is a timely need to investigate and assess data on the public’s knowledge and individual resilience with respect to the COVID-19 pandemic. Knowledge, perceived knowledge, perceived safety, and personal resilience may, individually or collectively, impact on the ability to recruit the public as an intrinsic component of disaster management. Accordingly, identifying ways to enhance them before or during periods of crises would support governments and health authorities in devising strategic and effective public health interventions, which may include measures such as appropriately informing the public, assessing impact of risk-communication strategies, or targeting information towards specific demographics.

During the COVID-19 pandemic, Magen David Adom (the Israeli emergency medical services - EMS) was involved in relaying information to the public concerning home-quarantine and testing symptomatic individuals for the virus. As these activities required the cooperation of the public, brief video tutorials were produced, in several languages, aimed at raising the knowledge, resilience, and perceived safety of the population. This EMS has previously acquired experience in training the public, as was demonstrated during security crises; training the public for providing first aid was found to decrease stress and anxiety [27].

In this study, we investigate the impact of a brief educational intervention on perceived knowledge, perceived safety, and the individual resilience of the population in relation to the COVID-19 outbreak. We also studied the overall trust of the public concerning the main responding authorities as well as practices concerning protective measures for COVID-19. We focus our analysis of the intervention specifically on the state of Israel, which has been afflicted by over ten thousand (10,095) confirmed cases and nearly 100 (95) deaths as of April 10, 2020 [6].

## 2. Materials and Methods

### 2.1. Study Design

Considering the importance of achieving understanding and resilience among the public concerning the current COVID-19 crisis, an intervention study was conducted in March 2020, in the midst of the outbreak. The study was based on exposing a representative sample of the Israeli population to a brief educational video tutorial that focused on varied aspects of the COVID-19 pandemic.

The study was based on three main components—filling a pre-intervention 10-minute self-report questionnaire, watching a 6-minute COVID-19 video tutorial, and filling an identical post-intervention questionnaire, immediately after watching the video.

The data was collected by the largest Israeli internet panel company that consists of over 100,000 panelists, representing all demographic and geographic sectors (http://www.ipanel.co.il). This internet panel provides an online platform that adheres to the stringent standards of the European Society for Opinion and Marketing Research (ESOMAR). A stratified sampling method was used, based on data published by the Israeli Central Bureau of Statistics to ensure representation of the varied characteristics of the Israeli population with regard to age, gender, level of religiosity, and geographic zones. The validity of internet panels has been presented and discussed extensively, and the use of this methodology has been rapidly growing [28].

### 2.2. Intervention

Based on interventional mapping (IM) [29], a designated 6-minute video tutorial was developed, targeted at the adult population, in order to impact on their knowledge and understanding of the COVID-19 pandemic and strengthen their capacities and competencies in coping with this disrupting crisis. The video was specifically directed to relay vital information to the public concerning the characteristics of COVID-19, its etiology, signs and symptoms, transmission routes, measures to combat infectivity, and guidelines concerning behavior during home-quarantine. The development of the educational tool was made by a team of experts in training programs. The original video tutorial was 13 min long, but following a pilot-test, was shortened to comply with the median engagement time that was found to be optimal for educational programs, i.e., up to 6 min [30]. The information in the video was delivered by a female physician, wearing uniform of the National Emergency Medical Services (Magen David Adom), which is perceived as a highly professional and trustworthy entity [31].

The research framework included literature review → design of educational video tutorial → pre-intervention assessment → watching the video tutorial → post-intervention assessment.

### 2.3. Participants

The sample size was determined based on OpenEpi (https://www.openepi.com/SampleSize), requiring 385 respondents. The study was conducted using a random internet sample of 501 participants who consented to participate in the research. To partake in the study, the participants had to confirm their willingness to voluntarily participate in the three components of the research (pre-post questionnaires and watching the tutorial). The data was collected anonymously, following approval of the Ethics Committee of the Tel Aviv University (number 0001121-1 from March 23, 2020).

### 2.4. The Study Tool 

The survey contained a brief introduction, which provided information on the background, objective, procedure, voluntary nature of participation, and declarations of anonymity and confidentiality.

The questionnaire consisted of seven parts, based on items and indices that were developed specifically for this study, except for the personal resilience, which was based on a validated tool developed by Connor and Davidson (Connor-Davidson Resilience Scale (CD-RISC)). The questionnaire was validated by five content experts and pilot tested on 20 individuals prior to its dissemination. The components of the questionnaire consisted of the following: (1) knowledge measured by 4 multiple choice questions encompassed characteristics of COVID-19, signs and symptoms, modes of virus’ transmission, and conditions that require home-quarantine; (2) perceived knowledge assessed by 7 items encompassing characteristics of COVID-19, signs and symptoms, transmission routes, proficiency of guidelines, and confidence in one’s knowledge (See Table A1 in the Appendix A). The items were ranked by a 5 point Likert scale, scaling from 1 = completely disagree to 5 = completely agree. Cronbach’s Alpha for the perceived knowledge index was α = 0.676 for pre intervention and α = 0.723 for post intervention; (3) perceived personal resilience portraying individual feelings of ability and strength in face of COVID-19 measured by 10 items ranked by a 5 point Likert scale, ranging from 1 = completely disagree, to 5 = completely agree. Cronbach’s Alpha for the perceived personal resilience index was α = 0.877 for pre intervention and α = 0.920 for post intervention; (4) perceived safety assessed by two items (“I feel protected from COVID-19 and I am concerned that I or a family member will contract COVID-19”) (see Table A2 in the Appendix A); the items were ranked by a 5 point Likert scale, ranging from 1 = completely disagree, to 5 = completely agree); (5) practices during COVID-19 pandemic were assessed by three yes/no questions relating to purchase of disinfectants, personal protective masks and storing food and water due to the crisis (see Table A3 in the Appendix A); (6) trust in four organizational entities (Police, National Ambulance Service, Ministry of Health, Health fund) in their response to COVID-19 measured by a 5 point Likert scale, scaling from 1 = Not trusting at all, to 5 = Very trusting (See Table A4 in the Appendix A). As the Cronbach’s Alpha was lower than 0.6, no indices were conducted; and 7) demographics, assessed by 10 items including gender, year of birth, place of residence, marital status, number of children, number of dependents, education, religion, degree of religiosity, and income.

### 2.5. Statistical Analysis

Descriptive statistics were used for describing the participants’ demographic characteristics (frequency, mean, and standard deviation). Paired sample t test was used to compare the differences between T1 (pre- intervention) and T2 (post- intervention) including calculation of the percent change ((T2 − T1)/T1 × 100%) and the effect size (mean differences/standard deviation of the differences). Pearson correlation tests were conducted to find the association between knowledge score, perceived knowledge, perceived safety, and personal resilience for COVID-19 at T2. The differences between these four variables according to those who purchase disinfectants compared to those that did not do so and demographic characteristics were analyzed using sample t test and Pearson correlation according to the type of variable. Repeated measure analysis was conducted to assess the interaction effect of sociodemographic characteristics on the differences between T1 to T2. All statistical analyses were performed using SPSS software version 25. P-values lower than 0.05 were considered to be statistically significant.

## 3. Results

The study evaluated 501 participants (presented in Table 1) before and after watching a brief video tutorial on COVID-19. The study group was 51% male, with a mean age 41 (± 14.8 SD), ranging from 18 to 70. Approximately 43% of participants characterized themselves as secular, while 21% identified as religious or ultra-religious. This is similar to the Israeli population overall. Most (72%) have over 12 years of education (vocational or academic degree), 27% earn less than the average income while 36% earn above it, and half of the participants identified as being in a relationship and having children.

As presented in Table 2, following the educational intervention all four variables that were examined in the study increased significantly compared to their level before watching the COVID-19 video: knowledge score (*p* < 0.001), perceived knowledge (*p* < 0.001), and personal resilience (*p* < 0.001) as well as concerning perceived safety (*p* < 0.001). The most substantial change was in perceived knowledge, presenting a nearly medium effect size (0.44), and an increase of 5.4% (from 4.1 to 4.3), while perceived safety presented a 12.4% increase with a lower effect size (0.28).

Repeated measures analysis was used to assess the interaction effect of socio-demographic and socio-economic characteristics on the differences between T1 to T2, including gender, age, level of education, level of income, and religiosity. The results presented that these socio-demographic and socio-economic variables did not moderate the effects of the intervention.

A significant positive correlation was found between perceived knowledge and personal resilience before and after the intervention (r = 0.278 *p* < 0.001; r = 0.238, *p* < 0.001 respectively), while no correlation was identified between the knowledge score and personal resilience before and after the intervention. Perceived safety was positively correlated with perceived knowledge only after the intervention (r = 0.147, *p* = 0.001) and with personal resilience both before and after the intervention (r = 402 *p* < 0.001; r = 0.469, *p* < 0.001 respectively), but not correlated with knowledge score. Furthermore, a minor correlation was found between the knowledge score and the perceived knowledge before and after the intervention (r = 0.121 *p* = 0.007; r = 0.098, *p* = 0.029 respectively). The correlations are presented in Table 3.

Approximately 60% of the respondents reported that they purchased disinfectants (liquid gel or wipes) and about 59% stocked water and food due to the COVID-19 crisis, while only 30% purchased protective masks. Examining the differences in levels of perceived safety (concern) between respondents that purchased or did not purchase disinfectants presented a significant difference (3.7 vs. 3.2 respectively; t = 3.89 *p* < 0.001). A similar trend was found concerning respondents that stockpiled food and water due to the crisis (3.7 vs. 3.3 respectively; t = 3.55 *p* < 0.001).

The results for mean level of trust in different entities were measured in T1 and are presented in Figure 1. A similarly high level of trust was displayed concerning the Ministry of Health and the Emergency Medical Services (4.05 and 4.03, respectively). The level of trust in both the Health Fund and the police force was found to be significantly lower.

Repeated measures analysis that was conducted to assess the variability between the levels of trust in the different entities presented significant differences (F_(1,500)_ = 200.74, *p* < 0.001). Using Bonferroni test for multiple comparisons presented significant differences (*p* < 0.001) between the levels of trust in all entities, except between the EMS and Ministry of Health (*p* > 0.05). See Table 4.

The association between the demographic characteristics and knowledge score, perceived knowledge, perceived safety and personal resilience during COVID-19 was investigated. Gender was found to be significant in regard to both perceived safety and personal resilience. Males compared to females reported higher levels of perceived safety (3.2 vs. 2.7 respectively; t = 4.95, *p* < 0.001) as well as personal resilience (3.7 vs. 3.5 respectively; t = 3.24, *p* = 0.001). Respondents with an above-average income compared to those who earn less than an average income reported higher levels of personal resilience (3.7 vs. 3.6, respectively; t = −2.02, *p* = 0.025) and higher knowledge scores (3.5 vs. 3.4 respectively; t = –2.40, *p* = 0.017). Perceived knowledge was higher among secular respondents compared to those who are traditional, religious, or ultra-religious (4.3 vs. 4.2 respectively; t = −2.06, *p* = 0.04). No association was found between participants’ age, years of education and number of children bellow 18 years old to any of the indices.

## 4. Discussion

Understanding the public’s active involvement in protective behavior for the purpose of mitigating the destructive consequences of an emergency is a crucial component of an effective response to adversity, particularly towards pandemics such as the current COVID-19 crisis [32]. It is, thus, of vital importance to identify educational mechanisms that are productive in enhancing such collaborative actions. Under this framework, the current study investigates the impacts of a brief educational intervention on the relationships between knowledge, perceived knowledge, individual resilience, and perceived safety concerning the COVID-19 outbreak.

The findings of this study demonstrate a significant overall increase in all four examined variables immediately following the educational intervention, to varying degrees of effect size. While previously educational interventions have been shown to be an effective method for improving knowledge, beliefs, and behavior [14,15,16], the results of this study expand the benefits of the educational intervention to include increased perceived knowledge, perceived safety, and individual resilience. The study identified several positive associations between examined variables, but particularly unexpected was the relationship between perceived knowledge and personal resilience, as this indicates that the perception of knowledge is more implicated in resilience, rather than actual knowledge. This observation supports the findings of Kupika et al. (2019), in which higher individual perceptions of knowledge in the context of climate change were shown to promote resilience [23].Previous studies presented that perceived risk (as suggested by Paton, Mclure and Burgelt, (2006) and Lopes (2000)) predict greater community resilience [16,17]. The positive associations observed in the current study between perceived safety and perceived knowledge and personal resilience may suggest that beyond risk perception, perceived safety may be a predictor of personal resilience, though this has to be more extensively studied to confirm or predict this hypothesis. Furthermore, resilience may be significantly impacted by both internal and external experiences; thus, it may be instable and susceptible to changes over time [33,34]. The positive associations that were found in this study are of utmost importance, as resilience has been shown to play a significant role in the decision-making process for uptake of control measures, behavioral engagement in disaster preparedness, and performance in stressful situations [24,25,35].

Public health officials should consider the utilization of an educational intervention such as that implemented in the current study as a tool to empower a population and to induce greater behavioral change. The outcomes of this study demonstrate a cost-effective and rapidly implementable intervention design, which can allow for the prompt and proactive dissemination of targeted and tailorable information to meet specific goals and objectives during emergencies and crisis.

Additional outcomes of this study reveal several demographic factors that are associated with the examined variables. For example, males reported higher perceived safety and personal resilience levels in comparison to their female counterparts. Perhaps the difference in resilience may have more to do with gender-based reporting patterns, and potentially follows similar previously observed trends as found in Bodas et al. (2019) where women were shown to more critically examine their capacities [36]. With respect to women’s lower perceived levels of safety, Bateman and Edwards (2002) suggest that women live at greater exposure to risk due to greater social vulnerability and hold larger caregiving roles in the household, and consequently have a lower perception of safety [37]. In order to further empower women, the facilitation of targeted supplementary education to this demographic group may contribute to improved individual resilience and perceived safety levels. As expected from previous studies [38,39], we observed that higher income participants report higher levels of personal resilience and score higher with respect to the knowledge score.

In our evaluation of trust in authorities and practices of the study population, we found that, on average, participants indicated higher levels of trust in the EMS and Ministry of Health with regard to responding to COVID-19, as compared to the police and Health Fund. A decrease in citizens’ trust in the police over time has been noted even before the COVID-19 pandemic [40], while trust in the Israeli EMS has proven to be more constant [41]. The elevated trust in the medical entities, such as the Ministry of Health and EMS that are responsible for the “fight” against the COVID-19 virus, could also be compared to the tendency of increased faith in the responding body in other types of struggles. For example, it has been seen that the trust in the military rises during periods of wars or conflicts more than in other public institutions [42]. Several of these differences have previously been acknowledged, such as in the findings of Zakrison, Hamel and Hwang (2004) [43], which indicated that homeless individuals had lower levels of trust in the police as compared to paramedics. Bodas and Peleg (2020) correspondingly report the relatively high levels of trust for the Ministry of Health among the Israeli population amidst the COVID-19 outbreak, as trust has repeatedly been acknowledged as a crucial component in public compliance during a pandemic [44,45,46]. Lastly, high levels of concern that the participant or a family member of the participant will become infected also served as a predictor of purchasing behavior adopted by the study population. Similar purchasing behavior was reported in Spain during the 2009 influenza A (H1N1) and was analogously affected by high concern for becoming infected [47]. As suggested by Sim et al. (2020), such behavior potentially functions as a coping response, which acts as a form of self- preservation and of one’s family [48]. Nonetheless, it cannot be ruled out that those who purchased disinfectants and masks and had higher levels of concern, were inherently more risk-aware individuals than the rest of the respondents, even before the COVID-19 pandemic erupted.

One limitation of this work is that it assesses situational knowledge, trust in the authorities, and practices only at the start of the crisis, even though these are likely to adjust as the outbreak progresses. This can be addressed by administering the study longitudinally. Nonetheless, as the compliance of the population is vital during the acute stages of the pandemic, there is great value in understanding their actual and perceived knowledge, trust and practices in this initial phase. Another limitation is that this study was conducted via the internet in order to ensure a quick response collection. While this methodology ensured a rapid turnover of information and provided a representative sample of the Israeli adult population, the study conclusions are limited to individuals who have access to a source of internet and high computing skills. An additional limitation is that the assessment of change was conducted immediately after the respondents viewed the video tutorial; it is important to assess the continued effect over time. As in all studies based on questionnaires, social desirability bias cannot be ruled out, including concerning declaration of levels of trust to varied first response entities. Nonetheless, as the study was conducted through an internet panel that is completely independent and separated from any response entity, this potential bias is most probably better controlled. Lastly, because this study was conducted in Hebrew, members of the Israeli population who are not fluent in the language were unable to participate.

## 5. Conclusions

Our findings have valuable implications as we show that an inexpensive and convenient short intervention can be an effective means for educating and empowering the public during the COVID-19 outbreak and should thus be considered as a component of the response in future outbreaks. As the Israeli public, despite its exposure to varied compromised security situations, has had no prior experience in coping with pandemics nor their consequences such as lockdowns or quarantines, the findings of the current study may be generalizable to other contexts. Future research should explore the longitudinal impacts of such educational interventions on changing behavioral practices and the uptake of control measures among participants.

## Figures and Tables

**Figure 1 ijerph-17-05971-f001:**
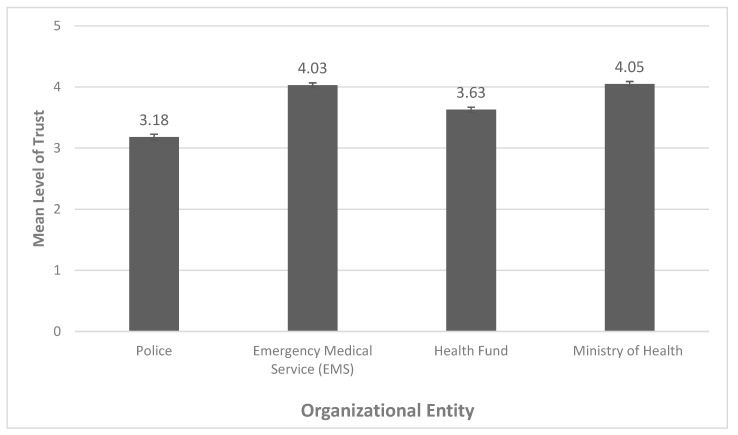
Mean levels of trust according to organizational entity. Errors bars are ± 1 SEM.

**Table 1 ijerph-17-05971-t001:** Characteristics of the study population (N = 501).

Characteristic	Number (Percentage) N = 501
Age	
18–21	53 (10.6%)
22–40	209 (41.7%)
41–60	176 (35.1%)
60–70	63 (12.6%)
**Gender**	
Male	253 (50.5%)
Female	248 (49.5%)
**Religiosity**	
Secular	214 (42.7%)
Traditional	184 (36.7%)
Religious	53 (10.8%)
Ultra-religious	48 (9.8%)
**Education** (in years)	
<12 years	139 (27.7%)
Vocational (non-academic)	102 (20.4%)
Academic	260 (51.9%)
**Area of residence**	
North (from Haifa)	132 (26.3%)
Central area	206 (40.1%)
Jerusalem area	55 (11%)
Southern area	108 (21.6%)
**Level of income ***	
Much below mean	56 (11.2%)
Below mean	77 (15.4%)
Mean	133 (26.5%)
Above mean	138 (27.5%)
Much above mean	44 (8.8%)
Refuse to answer	53 (10.6%)
**Marital status**	
In a relationship without children	107 (21.4%)
In a relationship with children	254 (50.7%)
Not in a relationship, without children	107 (21.4%)
Not in a relationship, with children	33 (6.6%)

* Mean income level is based on the Israel Central Bureau of Statistics.

**Table 2 ijerph-17-05971-t002:** Differences in knowledge, perceived knowledge, perceived safety, and personal resilience for COVID-19 between T1 and T2 (pre-post intervention).

Variable	T_1_	T_2_	% Change	Effect Size * (Cohen’s d) **	*p*-Value
Knowledge score	3.21 ± 0.78 3.0 (3.0–4.0)	3.46 ± 0.70 4.0 (3.0–4.0)	7.8%	0.34	<0.001
Knowledge perception	4.06 ± 0.51 4.0 (3.7–4.4)	4.28 ± 0.45 4.4 (4.0–4.6)	5.4%	0.44	<0.001
Perceived safety	2.58 ± 1.15 2.0 (2.0–3.0)	2.90 ± 1.16 3.0 (2.0–4.0)	12.4%	0.28	<0.001
Perceived personal resilience	3.51 ± 0.69 3.5 (3.1–4.0)	3.59 ± 0.76 3.7 (3.1–4.0)	2.5%	0.11	<0.001

Data are presented as mean ± standard deviation and median (Q25–Q75). *p*-value is based on paired sample t Test. Effect size *: 0.2-Small, 0.5-Medium, 0.8-Large. Cohen’s d **: appropriate effect size for the comparison between two means.

**Table 3 ijerph-17-05971-t003:** Pearson correlations between knowledge scores, perceived knowledge, perceived safety, and personal resilience for COVID-19 at T1 and T2.

Variable	Personal Resilience	Knowledge Score	Perceived Safety
**T1**			
Knowledge score	r = 0.062 *p* = 0.167		r = − 0.073 *p* = 0.103
Perceived knowledge	r = 0.278 *p* < 0.001	r = 0.121 *p* = 0.007	r = 0.082 *p* = 0.067
Personal resilience		r = −0.167 *p* = 0.501	r = 0.402 *p* < 0.001
**T2**			
	**Personal Resilience**	**Knowledge Score**	**Perceived Safety**
Knowledge score	r = −0.050 *p* = 0.261		r = −0.86 *p* = 0.054
Perceived knowledge	r = 0.238 *p* < 0.001	r = 0.098 *p* = 0.029	r = 0.147 *p* = 0.001
Personal resilience		r = − 0.050 *p* = 0.261	r = 0.469 *p* < 0.001

**Table 4 ijerph-17-05971-t004:** Results of Bonferroni test of the levels of trust in varied response entities.

Compared Entities	Significance
Police vs. Emergency Medical Services	*p* < 0.01
Police vs. Health Fund	*p* < 0.01
Police vs. Ministry of Health	*p* < 0.01
Emergency Medical Services vs. Health Fund	*p* < 0.01
Emergency Medical Services vs. Ministry of Health	*p* = 0.32
Health Fund vs. Ministry of Health	*p* < 0.01

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
