# Peer review of "Impact of a Brief Educational Intervention on Knowledge, Perceived Knowledge, Perceived Safety, and Resilience of the Public During COVID-19 Crisis"

_ijerph, 2020, doi:10.3390/ijerph17165971_

Round 1
Reviewer 1 Report
This is an interesting study and I appreciate the opportunity to read and provide my thoughts about it. My main recommendation is that the study can be published with a few modifications and further clarifications which I briefly outline below:
- First of all, it was not very clear to me what was the timing of pre- and post-treatment assessment of individuals’ knowledge and perceptions. If post-treatment assessment was performed immediately after treatment (within minutes), I wouldn’t at all be surprised that the treatment had some effect. I think more interesting and enlightening would be if authors have tested the lasting effect of the treatment after a few days or a week. The authors should at least discuss timing of treatment and its implications for findings.
- The authors choose to present results only from bivariate associations without properly accounting how individuals’ different socio-demographic and socio-economic variables explain the observed results of interventions. It is true that the last paragraph before the discussion section describes how individuals’ characteristics are associated with knowledge and perceptions but not how these characteristics moderate the effect of the intervention. For instance, were more educated individuals more likely to understand the content of intervention?
- The previous point relates to possible selection issues among individuals. For instance, those who purchased disinfectants and mask and have higher levels of concern are likely inherently more risk-aware individuals than the rest even before the COVID-19 pandemic started.
- Lastly, Israel seems to be a quite unique country in terms of its population’s experience and readiness to situations resembling the quarantine/lockdown. The authors might discuss how their results are generalisable to other contexts. Also, different levels of trust in police and emergency services are very interesting and authors could elaborate a bit more if these differences were also observed before the pandemic. It would be interesting to know, for instance, if during war periods trust in military increases more than trust in other public institutions.
Author Response
We would like to thank the reviewer for his/her comments. We appreciate the opportunity to revise the manuscript based on these wise comments and believe the article is now substantially improved. All the comments were adhered to, corrected and embedded in the updated revised manuscript.
Following are point by point responses to the reviewer’s comments.
This is an interesting study and I appreciate the opportunity to read and provide my thoughts about it. My main recommendation is that the study can be published with a few modifications and further clarifications which I briefly outline below:
- First of all, it was not very clear to me what was the timing of pre- and post-treatment assessment of individuals’ knowledge and perceptions. If post-treatment assessment was performed immediately after treatment (within minutes), I wouldn’t at all be surprised that the treatment had some effect. I think more interesting and enlightening would be if authors have tested the lasting effect of the treatment after a few days or a week. The authors should at least discuss timing of treatment and its implications for findings.
Response: The post-intervention assessment was performed immediately after the respondents watched the video tutorial. We added this time-frame to the methods section (see page 3, line 117), and discussed its implications on the findings in both the discussion section (page 10 line 277) and the limitations section (page 22 lines347-349)
2. The authors choose to present results only from bivariate associations without properly accounting how individuals’ different socio-demographic and socio-economic variables explain the observed results of interventions. It is true that the last paragraph before the discussion section describes how individuals’ characteristics are associated with knowledge and perceptions but not how these characteristics moderate the effect of the intervention. For instance, were more educated individuals more likely to understand the content of intervention?
Response: We added repeated measure analysis to assess the interaction effect of sociodemographic characteristics on the differences between T1 to T2. See page 6 lines 215-219.
3. The previous point relates to possible selection issues among individuals. For instance, those who purchased disinfectants and mask and have higher levels of concern are likely inherently more risk-aware individuals than the rest even before the COVID-19 pandemic started.
Response: We added this insight to the discussion section. See page 11 lines 336-338.
4. Lastly, Israel seems to be a quite unique country in terms of its population's experience and readiness to situations resembling the quarantine/lockdown. The authors might discuss how their results are generalisable to other contexts.
Response: The Israeli population, despite its exposure to security threats, did not have any prior experience in coping with pandemics nor their consequences, such as lockdowns or quarantines. Thus, the findings of the current study may be generalizable to other contexts. We added this insight to the text. See page 11 lines359-361.
Furthermore, different levels of trust in police and emergency services are very interesting and authors could elaborate a bit more if these differences were also observed before the pandemic. It would be interesting to know, for instance, if during war periods trust in military increases more than trust in other public institutions.
Response: There were different levels of trust in the police vs EMS prior to COVID-19 pandemic. Such differences were added to the discussion section. We also added information concerning increase in trust of the military during conflicts/security risks. See pages 10-11 lines 319-325
Reviewer 2 Report
Dear colleagues
thanks for this article highlighting a brief intervention on COVID-19-related behavioral change.
Some comments I have:
2.2 Intervention
Please describe the intervention in a bit more detail. Was it developed based on behavioral change techniques? Or with intervention mapping? How did you know which age groups it would be appropriate for? Why 6 minutes and not longer or shorter? Overall please describe the process of developing the intervention (which experts were involved, which theories was it based on, was it pilot-tested). In the discussion you write it was inexpensive to administer, how much would it cost to implement? Why do you assume that a person wearing the Magen David Adom uniform is automatically perceived as trustworthy? Please cite sources for this assumption.
2.3 Participants
random internet sample means what? If based on data from your census office, how did you reach the participants online? How did you ensure that the stratification made it representative, was their weighting for certain regions? How about inequalities in internet use between age groups? In short, we need a bit more information.
2.4 The Study Tool
Could you add a definition of "personal resilience"? I think this is also needed for the discussion later, as it seems difficult to measure changes in resilience after 6 minutes and one video.
3 Results
Could you state whether the sample distribution is similar to your larger population, e.g. is the ratio of secular to religious similar to the Israeli population overall? If not this should also be discussed as a limitation in the discussion section.
4 Discussion
I am missing a discussion of how socially desirable answers towards the level of trust may have been in a survey conducted by a national health organisation asking about other national organisation. Could you mention why you think these answers may still have been truthful as part of your researcher reflectivity?
Author Response
We would like to thank the reviewer for his/her comments. We appreciate the opportunity to revise the manuscript based on these wise comments and believe the article is now substantially improved. All the comments were adhered to, corrected and embedded in the updated revised manuscript.
Following are point by point responses to the reviewer’s comments.
Some comments I have:
2.2 Intervention
Please describe the intervention in a bit more detail. Was it developed based on behavioral change techniques? Or with intervention mapping? How did you know which age groups it would be appropriate for? Why 6 minutes and not longer or shorter? Overall please describe the process of developing the intervention (which experts were involved, which theories was it based on, was it pilot-tested).
Response: The process of developing the educational tool and intervention has been added to the methods section. See page 3 lines 128-131; lines 134-137.
In the discussion you write it was inexpensive to administer, how much would it cost to implement?
Response: As the development was made in-house by the EMS, based on experts and infrastructure that are available within the organization no external costs were needed. The dissemination was through social media so did not necessitate any costs.
Why do you assume that a person wearing the Magen David Adom uniform is automatically perceived as trustworthy? Please cite sources for this assumption.
Response: This is based on previous studies that reflected this finding. The reference to the source was added to the intervention section in the methods. See page 3 line 139.
2.3 Participants
random internet sample means what? If based on data from your census office, how did you reach the participants online? How did you ensure that the stratification made it representative, was their weighting for certain regions? How about inequalities in internet use between age groups? In short, we need a bit more information.
Response: We further elaborated on the study design as requested. See page 3 lines 119-125, and also added a reference as to the validity of internet panels.
2.4 The Study Tool
Could you add a definition of "personal resilience"? I think this is also needed for the discussion later, as it seems difficult to measure changes in resilience after 6 minutes and one video.
Response: The definition was added to the introduction section (see page 2 lines 76-77)
3 Results
Could you state whether the sample distribution is similar to your larger population, e.g. is the ratio of secular to religious similar to the Israeli population overall? If not this should also be discussed as a limitation in the discussion section
Response: The sample distribution is similar to the overall Israeli population, in line with the data published by the Israeli Central Bureau of Statistics. This was added to the study design section in the methods. See page 3 lines 121 124
4 Discussion
I am missing a discussion of how socially desirable answers towards the level of trust may have been in a survey conducted by a national health organisation asking about other national organisation. Could you mention why you think these answers may still have been truthful as part of your researcher reflectivity?
Response: We added a reference to this issue in the limitations section of the discussion. See page 11 lines 350-354.
Reviewer 3 Report
Line 36-37:Causality should follow chronological order. The outbreak announced by the WHO is based on data before March 11, 2020, but the expression of “this rapid growth” showed that the author here uses data after March 11, 2020(March 11 to May 23,2020) to explain the decision made by WHO on March 11, 2020, which does not conform to the usual thinking logic.
Line 132:5)is missing before ”the items” .
Line 148:better use the form
((T2-T1)/T1×100%)
Line 186:What indicators are used to measure the levels of concern, I did not find them in the questionnaire.
Line 195:The results of the Bonferroni test are recommended to be displayed in a table
In addition:
1.It is recommended to display the questionnaire introduced in 2.4 in the form of a table and place it in the appendix
2.According to the abstract and the introduction(the last paragraph), the purpose of this article is to investigate the impact of a brief educational interventions on KAP(knowledge, attitudes, and practices), perceived knowledge, perceived safety, and individual adaptability of people related to the COVID-19 outbreak. However, in the last three paragraphs of the result, the author did not compare the changes in the practices and attitudes before and after the educational intervention, which seems to go against the original intention of the paper.
3 relate to 2. The abstract and the last paragraph of introduction do not reflect the full content and results of the research which are needed to be reorganized.
4.It is recommended to add a research framework to the section of material and method, which can allow readers to understand the content of this article more clearly.
Author Response
We would like to thank the reviewer for his/her comments. We appreciate the opportunity to revise the manuscript based on these wise comments and believe the article is now substantially improved. All the comments were adhered to, corrected and embedded in the updated revised manuscript.
Following are point by point responses to the reviewer’s comments.
Line 36-37:Causality should follow chronological order. The outbreak announced by the WHO is based on data before March 11, 2020, but the expression of “this rapid growth” showed that the author here uses data after March 11, 2020(March 11 to May 23,2020) to explain the decision made by WHO on March 11, 2020, which does not conform to the usual thinking logic.
Response: This was corrected so that there is no causality. See page 1 lines 41-42
Line 132:5)is missing before ”the items” .
Response: This was corrected; we added the needed number. See page 4 line 173
Line 148:better use the form
((T2-T1)/T1×100%)
Response: This was corrected as advised. See page 5 line 190
Line 186:What indicators are used to measure the levels of concern, I did not find them in the questionnaire.
Response: The indicators are delineated in the study tool section in the methods; see page 4 lines 171-172). Furthermore, the full indices and questions were added as an annex to the document. See pages 15-18 lines 521-585.
Line 195:The results of the Bonferroni test are recommended to be displayed in a table
Response: The results were added as Table 4. See page 8 lines 255-256
In addition:
1.It is recommended to display the questionnaire introduced in 2.4 in the form of a table and place it in the appendix
Response: the questionnaire was added as an annex to the article. See pages 15-18 lines 521-585.
2.According to the abstract and the introduction(the last paragraph), the purpose of this article is to investigate the impact of a brief educational interventions on KAP(knowledge, attitudes, and practices), perceived knowledge, perceived safety, and individual adaptability of people related to the COVID-19 outbreak. However, in the last three paragraphs of the result, the author did not compare the changes in the practices and attitudes before and after the educational intervention, which seems to go against the original intention of the paper.
Response: Practices and attitudes (level of trust) were measured only once as we do not think it is possible to revise them during the intervention time (for example, if you did not buy masks, you are not able to do so during the pre-post intervention (due to their immediacy). Nonetheless, following the reviewer’s comments, we revised the abstract, the introduction and the results sections. See page 1 lines 21, 24-26; page 2 line 62, page 3 lines 104-105, page 8 line 242.
3 relate to 2. The abstract and the last paragraph of introduction do not reflect the full content and results of the research which are needed to be reorganized.
Response: As stated in the former bullet point, these have now been revised.
4.It is recommended to add a research framework to the section of material and method, which can allow readers to understand the content of this article more clearly.
Response: The research framework was added to the methods section. See page3 lines 140-142.
Reviewer 4 Report
The manuscript aims to explore the relationships between educational intervention and public’s knowledge, perceived knowledge, perceived safety and resilience during COVID-19 crisis. My opinion is that, although the topic is somehow interesting, the current version of the manuscript suffers from important problems that compromise its quality, precision and value.
Below please find my comments.
Comment 1:
when I read the Introduction, the logic is confused and the description is unclear.
First, many words were used to explain the compliance of public which is not the research object of this study. I suggest the authors directly present the roles of disaster education intervention at the first beginning of the section.
Second, many concepts were mentioned in this study, such as knowledge, attitude, practice, perceived knowledge, perceived safety, resilience. However, almost of the concepts are not explained. I suggest the authors explained these concepts in the section of Introduction with more details.
Third, the description of relationships among the variables is very fuzzy, such as P2 Line 51-58. What is the purpose of writing this passage? More important, what is your point? Just like my first impression after reading this article, it is more like data-driven study but lack of a theoretical basis. I suggest the authors describe the important relationships with clearer structure as well as more clearly propose your own ideas or hypotheses.
Comment 2:
In the section of Methods, I have some questions about the measurement.
First, as you mentioned “personal resilience portraying individual feelings of ability and strength in face of COVID-19 measured”, it is better to be defined as “perceived personal resilience”.
Second, I do not agree with the measurement of “attitude”, as you said it was measured by “evaluating trust in four organizational entities”. When we talk about attitude, it is usually followed by an object, such as “attitude toward preparedness behaviours”. Here, I think it seems more like talking about “trust in government”.
Third, why do you only show the reliability of two variables (perceived knowledge & personal resilience), how about others? Moreover, I also very care about the validity of these variables.
Fourth, since almost of the scales were developed specifically for this study, I suggest the authors present the items in a table.
Comment 3:
The structure of section of Results is also unclear. Some of the results (e.g. knowledge, perceived knowledge, perceived safety, and personal resilience) were analysed by a comparation between before and after intervention. While some (e.g. trust) was tested without comparation. Why? The same question also for the correlation analysis. Although in the part of Limitation, the authors provide some explanation, I suggest the authors restructure this part in Results for easier reading.
Besides, on P6 Line 178-179, you said “no substantial correlation was found between the knowledge score and the perceived knowledge (r=.098, p=.029)”, while on P5 Line 154, you mentioned “P-values lower than 0.05 were considered to be statistically significant”. This is contradictory.
Comment 4:
In the section of Discussion, I think some of the description is insufficiently rigorous. Such as on P8 Line 229-232, it was mentioned “the positive associations observed between perceived safety and each of the perceived knowledge and personal resilience, interestingly suggest that increases in perceived safety, rather than perceived risk (as suggested by Paton, Mclure and Burgelt, (2006) and Lopes (2000)) predict greater community resilience [18,19].” On one hand, it is hard to compare between different variables, such as personal resilience and community resilience. On the other hand, I only can see the positive correlation between perceived safety and personal resilience in this study, but I do not agree that you can judge the predictive effect based on current study.
Author Response
We would like to thank the reviewer for his/her comments. We appreciate the opportunity to revise the manuscript based on these wise comments and believe the article is now substantially improved. All the comments were adhered to, corrected and embedded in the updated revised manuscript.
Following are point by point responses to the reviewer’s comments.
The manuscript aims to explore the relationships between educational intervention and public’s knowledge, perceived knowledge, perceived safety and resilience during COVID-19 crisis. My opinion is that, although the topic is somehow interesting, the current version of the manuscript suffers from important problems that compromise its quality, precision and value.
Below please find my comments.
Comment 1:
when I read the Introduction, the logic is confused and the description is unclear.
First, many words were used to explain the compliance of public which is not the research object of this study. I suggest the authors directly present the roles of disaster education intervention at the first beginning of the section.
Response: In line with the reviewer’s comments, we deleted the reference to the topic of compliance. See page 3 lines 49-53.
Furthermore, as suggested by the reviewer, the topic of disaster education intervention was moved to the beginning of the section. See page 3 lines 52-58.
Second, many concepts were mentioned in this study, such as knowledge, attitude, practice, perceived knowledge, perceived safety, resilience. However, almost of the concepts are not explained. I suggest the authors explained these concepts in the section of Introduction with more details.
Response: The definitions of the concepts were added to the introduction. See page 3 lines 59-61, 67-69, 76-77.
Third, the description of relationships among the variables is very fuzzy, such as P2 Line 51-58. What is the purpose of writing this passage? More important, what is your point? Just like my first impression after reading this article, it is more like data-driven study but lack of a theoretical basis. I suggest the authors describe the important relationships with clearer structure as well as more clearly propose your own ideas or hypotheses.
Response: The description of relationships among the variables was added to the introduction. See page 3 lines 81-84.
Comment 2:
In the section of Methods, I have some questions about the measurement.
First, as you mentioned “personal resilience portraying individual feelings of ability and strength in face of COVID-19 measured”, it is better to be defined as “perceived personal resilience”.
Response: This was corrected as suggested. See page 4 lines 167, 170 and page 6, table 2
Second, I do not agree with the measurement of “attitude”, as you said it was measured by “evaluating trust in four organizational entities”. When we talk about attitude, it is usually followed by an object, such as “attitude toward preparedness behaviours”. Here, I think it seems more like talking about “trust in government”.
Response: The concept was revised as suggested by the reviewer. The term attitudes was deleted and replaced by trust in authorities. See page 2 lines 62-62; page 4 line 175; page 10 line 318; and page 11 line 341.
Third, why do you only show the reliability of two variables (perceived knowledge & personal resilience), how about others? Moreover, I also very care about the validity of these variables.
Response: Regarding the reliability, we calculated the Cronbach Alpha for the indices: 'Perceived Knowledge', 'Personal Resilience' and 'Trust'. Since the Cronbach alpha for 'Trust' was less than 0.6 we didn’t build an index and analyzed each of the questions separately. 'Perceived safety' was based on 2 questions only, therefore we didn’t build an index for this parameter. 'Practices' and 'Knowledge' measures were based on Yes/No questions – therefore no Cronbach Alpha was calculated. 'Perceived safety' was based on one question only. The results of the Cronbach Alpha tests are presented in the method section (under the sub-section: the study tool) for both T1 and T2 measurements; see page 4 lines 166-167, 170-171, 178-179.
Fourth, since almost of the scales were developed specifically for this study, I suggest the authors present the items in a table.
Response: All the items were added as an annex to the manuscript. See pages 15-18 lines 521-585.
Comment 3:
The structure of section of Results is also unclear. Some of the results (e.g. knowledge, perceived knowledge, perceived safety, and personal resilience) were analysed by a comparation between before and after intervention. While some (e.g. trust) was tested without comparation. Why? The same question also for the correlation analysis. Although in the part of Limitation, the authors provide some explanation, I suggest the authors restructure this part in Results for easier reading.
Response: We added to table 3 the correlation analysis for the pre- intervention period. See pages 6-7 line 234. The questions concerning trust in the 4 entities (Police, National Ambulance Service, Ministry of Health, Health fund) were asked only prior to the intervention.
Besides, on P6 Line 178-179, you said “no substantial correlation was found between the knowledge score and the perceived knowledge (r=.098, p=.029)”, while on P5 Line 154, you mentioned “P-values lower than 0.05 were considered to be statistically significant”. This is contradictory.
Response: What we meant is that the level of correlation is so low that it has no actual impact. Following the reviewer’s comment, and in order to ensure clarity, we rephrased the result to "minor correlation…". See page 6 line 230.
Comment 4:
In the section of Discussion, I think some of the description is insufficiently rigorous. Such as on P8 Line 229-232, it was mentioned “the positive associations observed between perceived safety and each of the perceived knowledge and personal resilience, interestingly suggest that increases in perceived safety, rather than perceived risk (as suggested by Paton, Mclure and Burgelt, (2006) and Lopes (2000)) predict greater community resilience [18,19].” On one hand, it is hard to compare between different variables, such as personal resilience and community resilience. On the other hand, I only can see the positive correlation between perceived safety and personal resilience in this study, but I do not agree that you can judge the predictive effect based on current study.
Response: We agree with the reviewer’s comment that we may have been “too quick to jump to a conclusion that cannot be proven by the findings of the current study”. We’ve revised the discussion accordingly. Specifically with this comment see page 10 lines 288-296.
Round 2
Reviewer 2 Report
Thank you for addressing my comments.
In Line 313 there is the word "Israel" which is either a typo or missing some context.
Reviewer 3 Report
This edition is much better than the previous one, I suggest accepting this article.
Reviewer 4 Report
Thank you very much for your revision. The current version of manuscript has a big improvement. However, please keep review the details in your manuscript, such as (1)the total population of religiosity in Table 1 is not equal to 501; (2)Page11-Line313?